Effects of the loss of estrogen on the heart’s hypertrophic response to chronic left ventricle volume overload in rats

Walsh-Wilkinson Elisabeth
Beaumont Catherine
Drolet Marie-Claude
Roy Ève-Marie
Le Houillier Charlie
Beaudoin Jonathan
Arsenault Marie
Couet Jacques jacques.couet@med.ulaval.ca
Groupe de Recherche sur les Valvulopathies, Centre de Recherche, Institut Universitaire de Cardiologie et de Pneumologie de Québec, Université Laval , Québec , Canada
Kennedy David
Electronic publication date: 2019 Oct 21
Publication date: 2019
Volume: 7
Electronic Location ID: e7924
Received 2019 Jul 9; Accepted 2019 Sep 20
Copyright: ©2019 Walsh-Wilkinson et al.
Copyright year: 2019
Copyright holder: Walsh-Wilkinson et al.
License: This is an open access article distributed under the terms of the Creative Commons Attribution License, which permits unrestricted use, distribution, reproduction and adaptation in any medium and for any purpose provided that it is properly attributed. For attribution, the original author(s), title, publication source (PeerJ) and either DOI or URL of the article must be cited.
License URL: https://creativecommons.org/licenses/by/4.0/

Keywords: Cardiac hypertrophy, Estrogens, Left ventricle, Aortic valve regurgitation, Rat, Ovariectomy, Volume overload, Sex dimorphism

Funding: Canadian Institutes of Health Research MOP-61818 MOP-106479 IUCPQ Foundation This work was supported by operating grants to Jacques Couet and Marie Arsenault from the Canadian Institutes of Health Research (MOP-61818 and MOP-106479) and the IUCPQ Foundation. The funders had no role in study design, data collection and analysis, decision to publish, or preparation of the manuscript.

==============================
Aortic valve regurgitation (AR) can result in heart failure from chronic overloading of the left ventricle (LV). Little is known of the role of estrogens in the LV responses to this condition. The aim of the study was to compare LV remodeling in female rats with severe AR in absence of estrogens by ovariectomy (Ovx). In a first study, we investigated over 6 months the development of hypertrophy in four groups of female Wistar rats: AR or sham-operated (sham) and Ovx or not. Ovx reduced normal heart growth. As expected, volume overload (VO) from AR resulted in significant LV dilation (42% and 32% increase LV end-diastolic diameter in intact and Ovx groups vs. their respective sham group; p < 0.0001). LV weight was also significantly and similarly increased in both AR groups (non-Ovx and Ovx). Increase in stroke volume or cardiac output and loss of systolic function were similar between AR intact and AR Ovx groups compared to sham. We then investigated what were the effects of 17beta-estradiol (E2; 0.03 mg/kg/day) treatment on the parameters studied in Ovx rats. Ovx reduced uterus weight by 85% and E2 treatment restored up to 65% of the normal weight. E2 also helped normalize heart size to normal values. On the other hand, it did not influence the extent of the hypertrophic response to AR. In fact, E2 treatment further reduced LV hypertrophy in AR Ovx rats (41% over Sham Ovx + E2). Systolic and diastolic functions parameters in AR Ovx + E2 were similar to intact AR animals. Ovx in sham rats had a significant effect on the LV gene expression of several hypertrophy markers. Atrial natriuretic peptide (Nppa) gene expression was reduced by Ovx in sham-operated females whereas brain natriuretic peptide (Nppb) expression was increased. Alpha (Myh6) and beta (Myh7) myosin heavy chain genes were also significantly modulated by Ovx in sham females. In AR rats, LV expression of both Nppa and Nppb genes were increased as expected. Ovx further increased it of AR rats for Nppa and did the opposite for Nppb. Interestingly, AR in Ovx rats had only minimal effects on Myh6 and Myh7 genes whereas they were modulated as expected for intact AR animals. In summary, loss of estrogens by Ovx in AR rats was not accompanied by a worsening of hypertrophy or cardiac function. Normal cardiac growth was reduced by Ovx in sham females but not the hypertrophic response to AR. On the other hand, Ovx had important effects on LV gene expression both in sham and AR female rats.

Introduction

Severe aortic valve regurgitation (AR) is a chronic disease that results in progressive left ventricular (LV) dilatation and eccentric hypertrophy. Although it is not the most frequent valvular disease in the Western world, it is estimated, based on the Framingham study, that 13% of the population suffers from some degree of AR (Singh et al., 1999). Moderate to severe secondary AR also occurs in a significant proportion of patients (5–10%) undergoing transcatheter aortic valve replacement (TAVR) (Leon et al., 2016). In poorer populations worldwide, mitral and/or aortic valve regurgitation is a frequent complication of acute rheumatic fever. Rheumatic valve diseases are still occurring at an alarming rate in low to middle-income countries and in poor communities, (Zühlke et al., 2017). Gender differences in cardiac remodeling, hypertrophy and clinical outcome have been identified in various cardiac diseases such as heart failure, hypertension, aortic valve stenosis and experimental models of pressure and volume overload (VO) (Blenck et al., 2016; Maric-Bilkan et al., 2016). The impact of valve regurgitation in women (mitral or aortic) has received little attention. Most clinical trials on chronic AR have focused mainly on male cohorts and gender specific adaptations of the LV in subjects suffering from chronic severe AR have not been investigated (Evangelista et al., 2005; Lin et al., 1994; Greenberg et al., 1988).

In prior studies using a rat model of chronic and severe AR, we showed that females had a hypertrophic response similar or stronger to LV VO than males (Drolet et al., 2006; Beaumont et al., 2017). The main difference was that the LV remodeling taking place in females was characterized by increased wall thickening resulting in a relatively preserved overall LV morphology compared to males (walls to chamber diameter ratio) (Beaumont et al., 2017). Moreover, AR females showed a tendency for better survival than AR males. More recently, we observed that loss of androgens in male rats was associated with reduced cardiac growth and decreased LV hypertrophic response to severe AR. This suggests a role of male sex hormones in the control of both physiological and pathological cardiac growth and hypertrophy (Beaumont et al., 2019).

It is believed that estrogens can provide a protection against cardiac hypertrophy (CH) development and evolution towards heart failure (HF). This has been demonstrated in animal models of pressure overload in female rats (Blenck et al., 2016). In the aorto-caval fistula (ACF) rat VO model, males progress faster toward HF and show poorer survival than females (Dent, Tappia & Dhalla, 2010a). Since sex steroids have a potent effect on differentiation, they could thus explain a large part of the sex dimorphism observed in cardiac hypertrophy caused by VO (Leinwand, 2003).

In the present study, we tested if LV remodeling and hypertrophy were influenced by estrogens in female rats suffering of chronic and severe AR. The effect of ovariectomy (Ovx) in females was assessed in order to answer this question. We also wanted to evaluate if Ovx was associated with decreased LV function and potentially worse survival in AR rats. In a second protocol, we studied the effects of 17beta-estradiol (E2) supplementation in AR Ovx female rats to investigate if it was associated with a reversal of the effects associated with the loss of estrogens.

Methods

Animals

Fifty-eight female Wistar rats (225 to 250 g) (Charles River, Saint-Constant, QC, Canada) were studied. Ovx females were purchased at the age of 9 weeks, one week after the procedure. Experimental groups were as followed: Sham-operated (Sham; n = 11), Ovx sham-operated (ShOvx; n = 10), AR; (n = 12) and Ovx AR (AROvx; n = 10). AR was induced at the age of 10 weeks as previously described by perforation of one or two aortic valve leaflets using a catheter via the right carotid and under echocardiographic guidance (Arsenault et al., 2002; Plante et al., 2003). Briefly, the right internal carotid artery was exposed and cannulated. Then, under continuous echocardiographic guidance, an 18-gauge epidural catheter was advanced toward the aortic valve in a retrograde manner. The sonographer guided the position and the advance of the catheter in the aorta while it was pushed through a leaflet of the aortic valve into the LV. Leaflet perforation was repeated if the severity of the regurgitant jet was considered insufficient by echocardiographic criteria. Sham-operated animals only had the ligation of their right carotid. Duration of the protocol was 6 months. Two additional Ovx groups were also studied, namely ShOvx + E2 (n = 6) and AROvx + E2 (n = 8). E2 (17beta-estradiol) was administered using a subcutaneous pellet implanted in the neck of the animal liberating 0.03 mg/kg/day for 3 months (Innovative research of America, Sarasota, FL). After this period, a second pellet was implanted to complete the protocol. The protocol was approved by the Université Laval’s Animal Protection Committee and followed the recommendations of the Canadian Council on Laboratory Animal Care.

Echocardiography

An echocardiographic exam (Philips HD11XE using an 12 MHz probe (S12)) was performed under isoflurane anesthesia (2%) two weeks after surgery to confirm AR severity and at the end of the protocol 26 weeks later, as previously described (Arsenault et al., 2013; Arsenault et al., 2002; Plante et al., 2003). The regurgitant fraction was estimated by the ratio of the forward systolic flow time–velocity integral (VTI) to the reversed diastolic flow VTI measured by pulsed Doppler in the thoracic descending aorta. At the end of the protocol, the heart and the lungs were harvested and weighed. Heart chambers were dissected, weighted and the LV was then quickly frozen in liquid nitrogen and kept at −80 C until further use.

Gene Expression Analysis by quantative RT-PCR

LV gene expression was quantified for 6 animals per group by quantitative RT-PCR as described elsewhere (Champetier et al., 2009). Pre-optimized primers were from QuantiTect (Qiagen) and IDT (Coralville, Iowa) (Table 1) and SsoAdvanced Universal SYBR Green Supermix (Bio Rad, Hercules, CA) was used. We used one pair of non-pre-optimized primers for the enoyl CoA hydratase, short chain 1 gene (Echs1) (5′-GCTTTCAGGGTGTCTTGATTTG-3′  and 5′-GAGCTATGCACTGCAGATAGT-3′; 95 bp transcript). We tested three different genes as possible housekeeping gene as control for this study. Cyclophilin A gene was chosen since it had the one most stable expression among the different groups.

Table 1 Name and symbol of all primer pairs used for gene expression analysis by quantitative RT-PCR.

The table also includes catalogue numbers (from IDT or Qiagen) and the size of the amplicon.

mRNA	Symbol	Catalog no.	Amplicon (bp)	
acyl CoA déshydrogenase, very long chain	Acadvl	Rn.PT.58.13279450	147	
carnitine palmitoyltransferase 2	Cpt2	QT00186473	150	
connective tissue growth factor	Ctgf	QT00182021	102	
cyclophilin A	Ppia	Rn.PT.39a,22214830	140	
cytochrome b-245 heavy chain (NOX2)	Nox2	Rn.PT.58.17749203	97	
2,4-dienoyl CoA reductase 1	Decr1	Rn.PT.58.44352482	120	
enolase 3, beta	Eno3	QT00180138	106	
estrogen related receptor, alpha	Erra	Rn.PT.58.5170310	111	
estrogen related receptor, gamma	Errg	Rn.PT.58.8028733	141	
fatty acid translocase/CD36	Fat/CD36	QT01702680	81	
hexokinase 1	Hk1	Rn.PT.58.8913174	108	
hydroxyacyl-CoA dehydrogenase	Hadh	Rn.PT.58.17867024	135	
hydroxyacyl-CoA dehydrogenase alpha	Hadha	Rn.PT.58.46222281	138	
lysyl oxidase, cardiac	Lox	Rn.PT.58.10677971	150	
myosin, heavy polypeptide 6, cardiac	Myh6	Rn.PT.58.8646063	150	
myosin, heavy polypeptide 7, cardiac	Myh7	Rn.PT.58.34623828	125	
NADPH oxidase 4	Nox4	Rn.PT.58.11992143	107	
natriuretic peptide precursor type A	Nppa	Rn.PT.58.5865224	79	
natriuretic peptide precursor type B	Nppb	Rn.PT.58.5595685	108	
phosphofructokinase	Pfkm	Rn.PT.58.17873275	122	
procollagen-1 alpha-1	Col1	Rn.PT.58.7562513	134	
procollagen-3 alpha-1	Col3	Rn.PT.58.11138874	100	
pyruvate dehydrogenase alpha 1	Pdha1	QT01830220	93	
pyruvate dehydrogenase kinase, isozyme 4	Pdk4	QT00189287	145	
retinoid X receptor gamma	Rxrg	Rn.PT.58.6519292	103	
solute carrier family 2 member 1	Glut1	QT00178024	85	
solute carrier family 2 member 4	Glut4	QT00175931	146	
superoxide dismutase 1, soluble	SOD1	Rn.PT.58,5432362	138	
superoxide dismutase 2, mitochondrial	SOD2	Rn.PT.58.7509049	107	
superoxide dismutase 3, extracellular	SOD3	QT00379358	92	

Statistical analysis

Results are presented as the mean and the standard error of the mean (SEM). Two-way ANOVA analysis was performed and Holm-Sidak’s post-test was used for comparison between the groups (Graph Pad Prism 8.1, San Diego, CA). A Student’s t-test was used when only two groups were compared. A p-value lower than 0.05 was considered significant.

Results

Effects of overiectomy on the hypertrophic response to chronic volume overload

AR was surgically induced in intact (non-Ovx) and Ovx Wistar female rats at the age of 10 weeks. The protocol had a duration of 26 weeks (6 months). All animals survived the duration of the protocol. In Table 2 are summarized the characteristics of the animals at the end of the protocol. Sham Ovx females were smaller and their heart lighter compared to Sham. Indexed heart weight for tibial length was also lower for Sham Ovx compared to Sham. When indexed for body weight, no difference was present. As expected, AR caused important increases in total heart weight as well as for the left ventricle and left atria. This increase was similar for both intact (non-Ovx) and Ovx animals (75% vs. 70%) as illustrated in Fig. 1. In order to confirm that Ovx resulted in a loss of sex hormones, we weighed the uterus, a tissue strongly dependant on estrogens. As expected, uterine weight was markedly decreased (84%) in both Ovx groups (sham and AR).

Table 2 Characteristics of the animals at the end of the protocol.

BW: body weight. Values are expressed as the mean ± SEM. Group comparisons were made using two-way ANOVA analysis and Holm-Sidak’s pos t-test.

Parameters	Sham (n = 11)	AR (n = 13)	Sham Ovx (n = 10)	AR Ovx (n = 10)	
Body weight, g	428 ± 15	418 ± 20	368 ± 11§	417 ± 14*	
Tibia, mm	51 ± 0.2	53 ± 0.3*	50 ± 0.3§	50 ± 0.2§	
Heart, mg	963 ± 20	1,685 ± 31*	765 ± 20§	1304 ± 25*§	
Heart/BW, mg/g	2.3 ± 0.1	4.1 ± 0.2*	2.1 ± 0.1	3.2 ± 0.1*§	
Heart/TL, mg/mm	18.8 ± 0.3	31.9 ± 0.6*	15.4 ± 0.4§	26.0 ± 0.5*§	
Left ventricle, mg	735 ± 17	1354 ± 25*	588 ± 13§	1,015 ± 25*§	
Left atria, mg	25 ± 3	47 ± 3*	18 ± 2	32 ± 2*	
Lungs, g	1.7 ± 0.1	3.2 ± 0.3*	2.4 ± 0.2	2.2 ± 0.3	
Uterus, mg	59 ± 4	59 ± 3	9 ± 1§	9 ± 1§	
Notes.

* p < 0.001 vs the respective sham group.

§ p < 0.05 vs non-Ovx group.

Figure 1 Ovariectomy does not modulate the hypertrophic response triggered by AR.

Results are expressed in arbitrary units (mean ± SEM) relative to their respective sham-operated group (fixed to 1). (A) Heart, (B) LV, Left ventricular weight, (C) EDD, end-diastolic diameter, (D) EF, ejection fraction, (E) SV, stroke volume and (F): CO, cardiac output. Calculated p values (Student’s T-test) are indicated for comparison between AR and AROvx groups. ∗∗∗: p < 0.001 compared to respective sham group (non-Ovx or Ovx).

As for the animal and heart characteristics described above, most echocardiographic parameters were significantly changed by AR (Table 3). AR severity was similar between both AR groups. LV end-diastolic diameter was smaller in AROvx animals compared to AR group. This was also the case for the stroke volume (SV) and the cardiac output (CO). Ejection fraction (EF; an index of systolic function) was reduced in both AR groups. Interestingly, loss of estrogens also associated with a reduced EF in Sham Ovx animals. The E wave, representing LV passive filling was significantly increased in AR animals compared to AR Ovx ones.

Table 3 Echocardiographic parameters of sham-operated animals at the end of the protocol.

Parameters	Sham (n = 11)	AR (n = 13)	Sham Ovx (n = 10)	AR Ovx (n = 10)	
AR severity, %	na	83 ± 4	na	78 ± 2	
EDD, mm	7.7 ± 0.1	10.9 ± 0.2	7.4 ± 0.1	9.8 ± 0.2*§	
ESD, mm	3.1 ± 0.1	6.5 ± 0.3*	3.6 ± 0.1	6.2 ± 0.3*	
SW, mm	1.1 ± 0.02	1.4 ± 0.05*	1.2 ± 0.04	1.4 ± 0.04*	
PW, mm	1.2 ± 0.03	1.8 ± 0.08*	1.4 ± 0.07	1.5 ± 0.06*	
EF, %	84 ± 2	65 ± 2*	75 ± 2§	61 ± 3*	
RWT, unitless	0.28 ± 0.005	0.26 ± 0.011	0.29 ± 0.010	0.27 ± 0.007	
SV, ml	0.29 ± 0.01	0.52 ± 0.04*	0.22 ± 0.01§	0.40 ± 0.02*§	
HR, bpm	386 ± 13	379 ± 9	348 ± 16	373 ± 11	
CO, ml/min	113 ± 3	187 ± 10*	79 ± 7§	148 ± 8*§	
E wave, cm/s	95 ± 4	109 ± 4	83 ± 4	86 ± 3§	
A wave, cm/s	61 ± 3	57 ± 2	59 ± 7	47 ± 2	
E wave slope	2992 ± 199	3379 ± 305	2098 ± 215§	2850 ± 135	
Notes.

EDD end-diastolic diameter

ESD end-systolic diameter

SW septum wall thickness

PW posterior wall thickness

EF ejection fraction

RWT relative wall thickness

SV stroke volume

HR heart rate

bpm beats per minute

CO cardiac output

na non applicable

Values are expressed as the mean ± SEM. Group comparisons were made using two-way ANOVA analysis and Holm-Sidak’s post-test.

* p < 0.05 vs. respective sham group.

§ p < 0.05 vs. non-Ovx group.

In Fig. 1, we illustrated variations of several parameters mentioned above in AR animals relative to their respective sham-operated group. As expected, AR caused important cardiac hypertrophy in both AR and AROvx animals compared to sham and this increase in heart weight was similar for both groups. A tendency for a greater increase in LV weight and LV end-diastolic diameter (EDD) caused by AR was recorded for the AR group compared to AROvx but this did not reach statistical significance (Fig. 1B–1C). Ejection fraction (EF), LV stroke volume (SV) and cardiac output (CO) were all modified by VO from AR but again, there was no difference from the hormonal status (AR group vs. AROvx) (Figs. 1D–1F).

Effects of E2 treatment on cardiac hypertrophy in AR Ovx females

We then studied the effects of E2 treatment in both Sham Ovx (ShOvx) and AR Ovx (ArOvx) rats. As summarized in Table 4, Ovx rats treated with E2 were still smaller than non-Ovx ones (see Table 2 for comparison). On the other hand, indexed heart weight was normalized suggesting that cardiac growth was not slowed in ShOvx + E2 rats. AR produced heart and LV hypertrophy, but relatively less than for intact non-Ovx or sham and Ovx animals (around 40% increase in AROvx + E2 compared to 70% for untreated AROvx; Table 2). Uterine weight was increased by E2 treatment to approximately 65% of these of non-Ovx females.

Table 4 Animal characteristics of Ovx animals treated with 17beta-estradiol (E2) at the end of the protocol.

BW: body weight and TL: tibial length. Values are expressed as the mean ± SEM. Group comparisons were made using Student’s T-test.

Parameters	ShOvx + E2 (n = 6)	AROvx + E2 (n = 8)	p-value	
Body weight, g	342 ± 8	320 ± 11	0.096	
Tibial length, mm	48 ± 0.3	48 ± 0.3	0.69	
Heart, mg	870 ± 26	1,223 ± 39	<0.0001	
Heart/BW, mg/g	2.5 ± 0.07	3.9 ± 0.10	<0.0001	
Heart/TL, mg/mm	18.0 ± 0.5	25.6 ± 0.7	<0.0001	
Left ventricle, mg	650 ± 20	973 ± 24	<0.0001	
Left atria, mg	21 ± 1	34 ± 3	<0.0001	
Lungs, mg	1.5 ± 0.1	1.6 ± 0.1	0.89	
Uterus, mg	37 ± 2.3	40 ± 3.1	0.55	

As illustrated in Fig. 2, E2 treatment partly normalized cardiac growth in Sham Ovx females. Heart and LV weights were also significantly increased in Sham Ovx rats receiving E2 compared to those untreated. Moreover, LV stroke volume (SV) and cardiac output (CO) were completely normalized by E2 treatment in sham females (Table 5 and Figs. 2E and 2F). Systolic function as evaluated by ejection fraction (EF) was unchanged be E2 treatment Fig. 2D.

Figure 2 Ovariectomy (Ovx) slows normal heart growth in Wistar female rats and E2 treatment partially reverses this effect.

Results are expressed as the ratio of the indicated parameter (mean ± SEM) compared to the mean of the same parameter for non-Ovx Sham females (set to 1). (A) Heart, (B) LV, Left ventricular weight, (C) EDD, end-diastolic diameter, (D) EF, ejection fraction, (E) SV, stroke volume and (F) CO, cardiac output. Calculated p values (Student’s T-test) are indicated for comparison between Sham (Ovx) and Sham (Ovx + E2) groups. ∗: p < 0.05 and ∗∗∗: p < 0.001 compared to non-Ovx sham female group.

LV gene expression modulation by estrogens

We then measured LV gene expression for several hypertrophy markers. Atrial natriuretic peptide (Nppa or Anp) and brain natriuretic (Nppb or Bnp) mRNA levels were both modulated by the loss of estrogens in Sham Ovx animals (Fig. 3A). Nppa levels were reduced by 60% whereas Nppb levels remained stable. Ovx modulated myosin heavy chain gene expression in a similar fashion as often observed in cardiac hypertrophy. Myosin heavy chain alpha (Myh6) gene expression was reduced by Ovx whereas myosin heavy chain beta (Myh7) was increased compared to non-Ovx Sham animals. Nppa mRNA levels were strongly increased in AR animals compared to corresponding Sham group; this raise being stronger in the AROvx group (Fig. 3B). We observed the opposite trend for Nppb mRNA levels, which were more strongly increased in AR females than in AROvx ones. A similar situation was observed for the expression of myosin heavy chain genes in AR rats. Myh6 gene expression was reduced and Myh7 increased by AR. Those modulations were less important in the AROvx group. Loss of estrogens lead to a decrease in procollagen 1 (Col1a1) and procollagen 3 (Col3) gene expression in sham-operated rats (Fig. 3C). Col1a1 and Col3 mRNA levels remained normal in AR animals (Fig. 3D) but were slightly more elevated in AROvx animals. The same was observed of mRNA levels of lysyl oxidase 1 gene (Lox) in AR groups. CTGF gene expression levels were unchanged by Ovx and were significantly increased by AR (Figs. 3C–3D). We then tested the expression of genes encoding transcription factors implicated in the control of myocardial energetics (Fig. 3E). Estrogen-related receptors (alpha and gamma) and retinoic X receptor gamma mRNA levels were measured in the LV of the animals. ERR alpha levels were reduced by Ovx in sham animals but not further by AR. Moreover, mRNA levels of these three transcription factor genes were significantly decreased by AR, but loss of estrogens restored these levels to normal (Fig. 3F).

Table 5 Echocardiographic parameters of Ovx animals treated with 17beta-estradiol (E2) at the end of the protocol.

Parameters	ShOvx + E2 (n = 6)	AROvx + E2 (n = 8)	p-value	
AR severity, %	na	66 ± 2	na	
EDD, mm	7.7 ± 0.1	9.7 ± 0.1	<0.0001	
ESD, mm	3.9 ± 0.1	5.6 ± 0.3	<0.0001	
SW, mm	1.1 ± 0.03	1.3 ± 0.05	0.0019	
PW, mm	1.3 ± 0.02	1.4 ± 0.10	<0.0001	
EF, %	74 ± 2	67 ± 3	0.073	
RWT, unitless	0.27 ± 0.007	0.26 ± 0.010	0.55	
SV, ml	0.30 ± 0.01	0.44 ± 0.02	<0.0001	
HR, bpm	362 ± 16	351 ± 10	0.56	
CO, ml/min	107 ± 5	153 ± 8	0.0005	
E wave, cm/s	90 ± 3	111 ± 5	0.0024	
A wave, cm/s	55 ± 2	67 ± 6	0.051	
E wave slope	2,700 ± 187	3,363 ± 255	0.0033	
Notes.

EDD end-diastolic diameter

ESD end-systolic diameter

SW septum wall thickness

PW posterior wall thickness

EF ejection fraction

RWT relative wall thickness

SV stroke volume

HR heart rate

bpm beats per minute

CO cardiac output

na non applicable

Values are expressed as the mean ± SEM. Group comparisons were made using Student’s T-test.

Figure 3 Evaluation by real-time quantitative RT-PCR of LV mRNA levels of genes encoding for hypertrophy markers (A, B and C), extracellular matrix genes (D and E) and transcription factors implicated in the control of myocardial energetics (F and G).

The results are reported as the mean ± SEM (n = 6/gr.) relative to non-Ovx Sham group (A, D and F) or to respective Sham group (non-Ovx (Blue) or Ovx (Orange)) (B, C, E and G). Messenger RNA levels of the respective sham group were normalized to 1 and are represented by the dotted line. *: p < 0.05, **: p < 0.01 and ***: p < 0.001 vs. respective sham group. §: p < 0.05 and §§§: p < 0.001 between indicated groups.

We reported previously that female AR rats unlike males, kept a relatively normal transcriptional profile of many genes related to myocardial energetics (Beaumont et al., 2017). Here, we were interested to see if loss of estrogens would induce pertubation to this. We thus tested a number of genes related to fatty acids beta-oxidation and glycolysis. In addition, we measured LV mRNA levels of various genes associated to reactive oxygen species (ROS) metabolism. As illustrated in Fig. 4, loss of estrogens via Ovx had very little effects on LV expression of various genes implicated in myocardial energetics except for one, Pdk4 (Figs. 4A and 4B). AR reduced the expression of a number of genes namely Acadvl, Decr1, Hadh, HadhA, Eno3 and Pdk4. Loss of estrogens did not further modulated those genes (AROvx rats). Among the genes related to ROS metabolism, we observed that NADPH oxidase 4 (Nox4) expression was significantly reduced by Ovx. This was reversed by AR. AR up-regulated Nox2 in both intact and Ovx females.

Figure 4 Genes implicated in energetics (A and B) and reactive oxygen species metabolism (C) are not modulated by the loss of estrogens in ShamOvx and AROvx rats.

The results are reported as the mean ± SEM (n = 6/gr.) relative to non-Ovx Sham group. Messenger RNA levels of non-Ovx Sham group were normalized to 1 and are represented by the dotted line. *: p < 0.05 and **: p < 0.01 vs. non-Ovx sham group.

We then studied if E2 treatment of Ovx animals helped restore changes observed in natriuretic peptides (Nppb or Bnp) and myosin heavy chains (Myh6 and Myh7) gene expression. Interestingly, E2 helped normalize Nppa and Myh7 expression in Sham animals (Fig. 5). Decreased Myh6 gene expression in Sham Ovx females was not normalized by E2, however. E2 treatment had no effect on gene expression levels in AR animals.

Figure 5 Evaluation by real-time quantitative RT-PCR of LV mRNA levels of genes encoding for hypertrophy markers in Sham Ovx and AR Ovx rats receiving (orange) or not (blue) 17beta-estradiol (E2) replacement.

(A) Nppa, (B) Nppb, (C) Myh6 and (D) Myh7. The results are reported as the mean ± SEM (n = 6/gr.) relative to non-Ovx Sham group (set to 1; dotted line). *: p < 0.05, **: p < 0.01 and between ***: p < 0.001 indicated groups. §§: p < 0.01 and §§§: p < 0.001 vs. non-ovx Sham group.

Discussion

In this study, we observed that loss of estrogens by ovariectomy (Ovx) two weeks before AR induction had relatively little effects on the extent of the cardiac response to a LV VO, at least at the macroscopic level. Ovx resulted in slower cardiac growth in Sham female rats during the 6 months that lasted the protocol. This was partly reversed by 17beta-estradiol treatment. On the other hand, LV hypertrophy caused by severe VO from AR was similar in all AR groups, non-Ovx, Ovx and Ovx receiving E2. We had previously shown that LV remodeling from AR in this model involved similar LV dilation in rats of both sexes but more wall thickening in females. This resulted in AR females in maintained LV relative wall thickness (RWT) although significant hypertrophy was present Beaumont et al. (2017). RWT also remained stable in sham-operated females after Ovx as well as in all AR groups. Our results suggest that loss of estrogens seems to clearly influence more cardiac normal growth than the response to chronic and severe VO in the AR rat model.

The roles of estrogens in pathological cardiac hypertrophy has been studied mostly in pressure overload animal models. It received less attention in VO situations such as in valve regurgitation models or in the aortocaval fistula (ACF) model. ACF is a model of global cardiac VO model. It is less relevant from a clinical standpoint but it remains the most studied pre-clinical VO model. Female ACF rats were shown to develop less hypertrophy, to evolve more slowly towards heart failure and to display better overall survival than males (Gardner, Brower & Janicki, 2002). This advantage over males was dependant on estrogens as ovariectomy reversed these benefits (Brower, Gardner & Janicki, 2003). Dent and collaborators characterized this ACF model further and showed that 17beta-estradiol could help normalize the effects of ovariectomy (Dent, Tappia & Dhalla, 2010b). Some discrepencies seem to exist between findings described in the present study in AR rats and those reported previously in the ACF model. A few differences have to be highlighted between these models. In the ACF model studies, evolution towards heart failure was documented (at least in males) whereas in the AR model, overt heart failure symptoms are a rare occurrence. In fact, most of the deaths are sudden happening during the active period of the animals during the night (Arsenault et al., 2013; Lachance et al., 2009; Plante et al., 2008). Since ACF is a global form of VO targeting the right heart first, it is likely that than lungs become seriously affected sooner, which leads to heart failure.

More than a decade ago, we had reported that Ovx was not associated with major effects in AR females (Drolet et al., 2006). More recently, we showed that LV dilation caused by AR had similarities between males and female rats, but that the expression profile of many genes involved in myocardial energetics was strongly modulated in males but not in females (Beaumont et al., 2017). This suggested that AR females could probably keep a relatively normal myocardial energy metabolism or at least, a better flexibility in energy substrate use even in a situation of pathological hypertrophy. In addition, myocardial capillaries density in AR females was not decreased as in males suggesting better oxygen and nutrients availability for surrounding cardiac myocytes. Removing androgens by orchiectomy (Ocx) in AR males reversed some of these sex differences. As observed in the present study for females, normal cardiac growth in male rats was also dependant on the presence of sex hormones. Unlike for Ovx females, hypertrophic response to severe VO was clearly decreased in Ocx AR males (Beaumont et al., 2019). Here, we observed that estrogens had minimal effects if any, on the LV remodeling taking place after AR induction. LV dilation, increase in stroke volume and cardiac output were similar in both non-Ovx and Ovx AR groups. This suggests that the hypertrophic response to a similar and direct LV pathological stress such as with AR required a similar myocardial adaptations to accommodate the additional regurgitating blood to pump. This was not influenced by the hormonal status. Moreover, this observation suggests that the female sex irrespective of estrogens can provide benefits in this rat model. It is also possible that imprinting of estrogens from the early life of the animals still remains.

The steroid hormone 17beta-estradiol is a key player in many biological processes, such as reproduction, development, metabolism, cell proliferation and differentiation (Deroo & Korach, 2006). Estrogens are implicated in the regulation of many genes and signaling pathways via genomic and non-genomic actions. E2 can bind and mediate its actions via the estrogen receptors (ER) ERalpha and ERbeta, which can then act as transcription factors for specific sets of genes. Membrane-associated receptors such as GPER and ERs, can also be activated by E2, resulting in the modulation of cytoplasmic signalling cascades and ultimately regulation of target genes (Murphy, 2011). Post-natal heart growth occurs via cardiomyocytes hypertrophy since these cells are post-mitotic. Activation of the ERalpha is required for post-natal heart growth in healthy Ovx mice (C57Bl6/J strain) receiving E2, suggesting a central role for this receptor (Kararigas, Nguyen & Jarry, 2014). ERbeta does not seem to be involved in normal heart growth in female mice but is believed to be implicated in the protection of the heart during a pathological stress (Mahmoodzadeh & Dworatzek, 2019). Estrogens can produce effects on the heart of males and females since ERs are present in the myocardium of both sexes. Estrogens as a potential therapy in men with cardiac diseases have received less attention than for women. E2 has been shown to rescue male mice with heart failure from transverse aortic constriction (a pressure overload model) via in part the ERbeta receptor and GPER (Iorga et al., 2016; Iorga et al., 2018). In the ACF model, estrogen therapy in males was able to reduce the hypertrophic response to the volume overload (Gardner et al., 2009). It would be interesting in the future, to investigate E2 effects in male AR animals in order to know if their effect could be more beneficial than in females.

Pathological cardiac hypertrophy is associated with an important remodelling of the myocardial structure, a consequence of cardiomyocyte size increase and extracellular matrix rearrangement. Neurohormonal factors as well as mechanic stress cause alterations in myocardial gene expression including the reactivation of the fetal gene program (Taegtmeyer, Sen & Vela, 2010). This feature is common to a variety of pathological conditions including ischemia, atrophy, hypoxia, diabetes in addition of hypertrophy. This return to the fetal gene program has long been considered detrimental, whereas others have suggested that it protects the heart against irreversible impairment and cell death. Genes associated with the fetal gene program include atrial and brain natriuretic peptide (Nppa and Nppb), contractile protein beta-myosin heavy chain (beta-MHC or Myh7) and early response genes such as c-myc and c-fos among many others. This reactivation of the fetal gene program in the stressed heart is accompanied with the down-regulation of the adult gene program (Rajabi et al., 2007). Here, we observed that Ovx was associated with the modulation of several genes associated with the fetal program in female rats. LV atrial natriuretic peptide (Nppa) and contractile protein beta-myosin heavy chain (Myh6 and Myh7) were all significantly modulated by the loss of estrogens. This was also true for other genes often modulated in cardiac hypertrophy such as collagen genes (Col1a1 and Col3), ERRalpha, Pdh4 and Nox4. This modulation was not associated directly with an inhibition or a reactivation of the fetal gene program. For instance, Myh6 and Myh7 LV genes in ShamOvx rats followed the usual pattern associated with pathological hypertrophy. On the other hand, Nppa was down-regulated and Nppb expression remained unchanged. If Nppa and Myh7 gene expression was normalized by E2, it was not the case for Myh6.

We reported previously that genes associated with myocardial energetics were strongly modulated in AR male rats and that a clear sex dimorphism was present when compared to females (Arsenault et al., 2013; Beaumont et al., 2017). Pathological LV hypertrophy is usually associated with a shift towards glucose use and glycolysis instead of the preferred beta-oxidation of fatty acids (FAO). We observed this shift in our model in males and this correlated with down-regulation of many genes involved in FAO (Arsenault et al., 2013; Lachance et al., 2014). Although, several FAO genes were down-regulated mildly in female AR rats compared to normal controls, the overall transcriptional profile remained near normal suggesting that they probably maintained a better energy substrates flexibility than males (Beaumont et al., 2017). Loss of androgens in males helps normalize this general down-regulation of FAO genes (Beaumont et al., 2019). We were thus interested to see if loss of estrogens would impact negatively AR females, which was not case as observed here. This suggests that androgens are probably more implicated in this control of myocardial energetics in pathological hypertrophy. As mentioned, the better angiogenic response during myocardial remodeling in females is probably a important contributing factor to the maintenance of relatively normal energetics.

The results obtained in the present study on female AR rats in conjunction to those we recently reported in males demonstrate that sex hormones are not the sole factors intervening in the LV hypertrophic response (Beaumont et al., 2019). Both androgens and estrogens are important for normal cardiac growth. Loss of estrogens by Ovx slows down cardiac growth and E2 treatment helps reverse this effect. Levels of LV hypertrophy are equivalent between AR males and females. Loss of testosterone reduces the extent of LV hypertrophy in AR rats whereas loss of E2 has relatively little effects. In addition, Ovx in AR rats is not associated with a worse transcriptional profile of genes normally regulated in cardiac hypertrophy. In fact, expression of several hypertrophy markers such as myosin heavy chain genes (Myh6 and Myh7) was in part normalized by Ovx in AR females as well as for mRNA levels of Nppb, Err alpha and Err gamma. Again, loss of androgens seemed to provide some benefits to males on this aspect whereas estrogens are mainly neutral in females (Beaumont et al., 2017; Beaumont et al., 2019). In summary, we did not identify clear negative impact of the loss of estrogens in AR female rats in a chronic setting. Sexual dimorphism in the response to VO seems to rely more on the effects of androgens in males. It is also possible that the influence of sex hormones before gonadectomy is still imprinted later in the life of the animals. Finally, effects of sex chromosomes and the genes they harbor, should not be excluded.

We want to point out several limitations in this study. In the second part of the study where Ovx rats received E2, estimated severity of AR was lower (66% in AR Ovx + E2 vs. around 80% for AR and AR Ovx groups) and so was the hypertrophy relative to the respective sham group (41% in AR Ovx + E2 vs. around 70–75% for AR and AR Ovx groups). On the other hand, if one considers the indexed heart weight gain between AR Ovx rats receiving or not E2, the hypertrophic response was similar. We did not measure circulating E2 levels and we used uterine weight as a surrogate of estrogen action. The dosage of E2 reversed about two thirds of the expected uterus weight suggesting that it was probably a little low or that type of delivery could not reproduce the natural situation. This could explain in part, why cardiac growth was not restored to normal levels. Obviously, continuous release of E2 does not reproduce naturally occurring circadian rhythm of production and release of sex hormones in the body. In addition, E2 treatment alone cannot restore completely other possible hormonal imbalances created by Ovx. They too, may probably contribute to observations made in this study. Finally, Ovx was performed in young animals, which does not translate well to the situation of post-menopausal and older patients.

Conclusion

In conclusion, we showed that loss of estrogens was not associated with important effects on the hypertrophic response to severe and chronic aortic valve regurgitation in female Wistar rats.

Supplemental Information

Supplemental Information 1 Raw data

Click here for additional data file.

Additional Information and Declarations

Competing Interests

Author Contributions

Animal Ethics

Data Availability

The authors declare there are no competing interests.

Elisabeth Walsh-Wilkinson and Catherine Beaumont performed the experiments, analyzed the data, prepared figures and/or tables, authored or reviewed drafts of the paper, approved the final draft.

Marie-Claude Drolet, Ève-Marie Roy and Charlie Le Houillier performed the experiments, analyzed the data, prepared figures and/or tables, approved the final draft.

Jonathan Beaudoin performed the experiments, authored or reviewed drafts of the paper, approved the final draft.

Marie Arsenault conceived and designed the experiments, authored or reviewed drafts of the paper, approved the final draft.

Jacques Couet conceived and designed the experiments, analyzed the data, prepared figures and/or tables, authored or reviewed drafts of the paper, approved the final draft.

The following information was supplied relating to ethical approvals (i.e., approving body and any reference numbers):

The protocol was approved by the Université Laval’s Animal Protection Committee and followed the recommendations of the Canadian Council on Laboratory Animal Care.

The following information was supplied regarding data availability:

Raw data for the results illustrated in the tables and figures of this article are available as a Supplemental File.

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
