# Peer review of "Effects of the loss of estrogen on the heart’s hypertrophic response to chronic left ventricle volume overload in rats"

_PeerJ, doi:10.7717/peerj.7924_

## Round 0.1 · original submission · Major Revisions

Your paper was reviewed by 2 independent experts in this field and there are several revisions which will be needed. The Reviewers have raised several issues related to the Basic Reporting, Experimental Design and Validity of the findings which should be thoroughly addressed in any subsequent revisions including clearly acknowledging and delineating any additional Limitations. Please also address each of the enumerated “Comments to authors” in your revision.

Reviewer 1 ·

Basic reporting

English definitely needs to be revised. Typos, spelling mistakes, missing spacing and non-meaningful sentences should be corrected

May abbreviations are using before being spelled out and explained

Experimental design

OVX and AR surgical procedures are not detailed in the methods sections. These details should be added.

Since multiple surgical procedures were performed in the current study, each time "sham operation " is mentioned, this needs to be clarifies: sham relative to which surgery

Were placebo pellets administered in control groups for E2-supplemented animals?

What is E2 levels in E2 supplemented animals and their control? These measurements should be included

Rationale for looking at genes expression of specific genes is not clear?

Explain why "Cyclophilin A" was used as a housekeeping gene? and not more commonly used ones?

Validity of the findings

In figure 1: It is not clear what stars are referring to in these 6 panels. The legend states that this is relative to sham groups, but sham group individual results are not displayed in the figure! It is fundamental to include this data set as a third group for each panel. Although the average would be one, still showing individual data will show variability.

It is not clear what is each p value included above panels refer to? and statistical test?

The type of statistical comparison tests used in figure 1 should be included in the legend

The authors relate their OVX effects as an effect ovarian hormonal depletion, which may not be true. OVX is associated with other hormonal differences so authors should be more accurate in describing their findings especially given that E2 supplementation did not restore all OVX-induced changes

Additional comments

Group codes are not explained in the abstract so it is hard to understand and follow the abstract

Reviewer 2 ·

Basic reporting

The study and data are well presented. Limitations are addressed by the authors. Overall well written with sufficient literature review. I have recommended that they include the effects of EST on cardiac remodeling in males as well, as those studies have been performed by others.

Experimental design

There are several limitations in the design, but the authors have addressed them in the discussion. Details of the ultrasound conditions should be included. I have included these in my comments below.

Validity of the findings

The authors do not overstate their conclusions and give a good discussion as to why their results in the AR model differ from published studies utilizing the ACF VO model.

Additional comments

Couet and colleagues present a study showing the effects of ovariectomy (OVX) and estrogen (EST) replacement on the cardiac function and remodeling response of wistar rats to chronic aortic regurgitation (AR). The study and data are well presented. Limitations are addressed by the authors. They show that OVX reduced the hypertrophic response to AR resulting but did not lead to further dysfunction when compared to intact rats. Estrogen replacement restored the hypertrophic response. Measures of ECM, metabolism, and fetal reprogramming were normalized by EST treatment.

Comments:
Define CH in abstract.

Differences in AR severity between the EST treated study and the OVX study make it difficult to compare, but the authors acknowledge this as a limitation.

Has anyone noted a difference in remodeling response between Wistar and Sprague Dawley rats in response to AR of ACF VO?

For Echo method, include a brief description of the anesthetic/sedation and ultrasound equip used.

Table 2: Have others shown a decrease in BW in response to OVX in female wistar? Were the animals pair fed?

Table 3: what EF do the Sham group have? The FS is very high. Also, the strongest effect on CO is due to OVX and not AR. Do the authors have a theory as to why? I do not believe that change occurs in Sprague Dawley females.

Fig 1 &2: recommend including x-axis bar labels in the top row as well.

Discussion page 8: define Ocx.

Est has been shown to reduce VO remodeling in male animals. Authors should include that information.

Fig 4 &5: recommend using hashed bar for OVX (Fig 4) and E2 (Fig 5) groups to aid in gray scale reproduction.

---

## Round 0.2 · Minor Revisions

Please include your response to Reviewer 1's query on E2 supplementation (i.e. E2 levels not available and use of uterine weight as a surrogate of estrogen action) in the Discussion/limitations section.

---

## Round 0.3 · accepted · Accept

Thank you for the opportunity to review this interesting work.